# Design of Auto-Tuning Nonlinear PID Tracking Speed Control for Electric Vehicle with Uncertainty Consideration

Mohamed A. Shamseldin [ID]

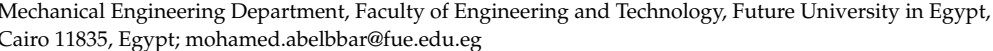

Mechanical Engineering Department, Faculty of Engineering and Technology, Future University in Egypt, Cairo 11835, Egypt; mohamed.abelbbar@fue.edu.eg

**Abstract:** This study presents a new auto-tuning nonlinear PID controller for a nonlinear electric vehicle (EV) model. The purpose of the proposed control was to achieve two aims. The first aim was to enhance the dynamic performance of the EV regarding internal and external disturbances. The second aim was to minimize the power consumption of the EV. To ensure that these aims were achieved, two famous controllers were implemented. The first was the PID controller based on the COVID-19 optimization. The second was the nonlinear PID (NPID) optimized controller, also using the COVID-19 optimization. Several driving cycles were executed to compare their dynamic performance and the power consumption. The results showed that the auto-tuning NPID had a smooth dynamic response, with a minimum rise and settling time compared to other control techniques (PID and NPID controllers). Moreover, it achieved low continuous power consumption throughout the driving cycles.

**Keywords:** electric vehicle (EV); adaptive control; nonlinear PID (NPID); COVID-19 optimization; model reference adaptive systems (MRAS)





## 1. Introduction

The use of electric vehicle (EV) technology is required to mitigate escalating environmental issues and to decrease the demand for fossil fuel resources [1]. In recent years, EVs have become increasingly popular due to their high efficiency, low maintenance requirements, and simple operations [2–4]. Urban cities now have improved sustainability and a significant reduction in pollution because of the growing EV trend. The performance of an EV as a whole is significantly influenced by its propulsion system. Industrial and academic researchers have mainly concentrated on creating controls for the electric vehicle's drivetrain [5,6]. The two most important aspects, efficient performance and desirable energy management, call for thorough and targeted research. The controller should deliver the fastest possible speed while consuming the least amount of energy [3]. Fluctuating road conditions, motor characteristics, and outside disturbances make the EV system highly nonlinear, time-dependent, and uncertain. As a result, it is difficult to build a controller that completely removes external disturbances and manages uncertainties with few control signals [7–9].

Due to their simplicity and ease of tuning, conventional PID controllers are frequently used in a variety of industrial applications [10]. However, they do not guarantee desired dynamic performance, and do not operate effectively under a variety of operating conditions with their self-tuning capabilities [11]. Due to windup, a PID controller produces a strong control signal, which causes it to overshoot and increase as the accumulated error is unwound (compensated by errors in the other direction), and the differentiator causes noise amplification [12,13]. Currently, there is no definite method to select the proper parameters. Hence, several optimization techniques can be used to solve this problem, such as genetic algorithm (GA), particle swarm optimization (PSO) [14], backtracking search algorithm (BSA), bee colony optimization (BCA), harmony search (HS), ant colony, differential evolution (DE), and COVID-19 optimization [14–17].

Artificial intelligence (AI)-based controllers have gained importance due to their satisfactory performance in various motor control applications, including speed assessment and torque ripple minimization [18–20]. However, AI-based controllers suffer from drawbacks, such as large data requirements, extended learning, and long training durations [21–23].

In this study, we concentrated on the nonlinear PID (NPID) controller, which has drawn a lot of interest from scholars over the past 20 years. NPID control is used by two main groups of applications. Only nonlinear systems that are controlled by NPID fall under the first category. NPID control is used in the second category, which deals with basic linear systems used to improve performance that are not feasible with linear PID control, such as decreased overshoot, decreased rise time for the step or rapid command input. NPID control improves following accuracy, and is utilized to account for nonlinearity and disturbances in the system [24–27].

The proposed improved NPID controller contains two portions. The first portion is a segment-bounded nonlinear gain $K_n(e)$, while the second portion is a linear fixed-gain PID controller ($K_p$, $K_i$ and $K_d$). The nonlinear gain $K_n(e)$ is a segment-constrained function of the error e(t). Previous research considered the nonlinear gain $K_n(e)$ as one scalar value. The novelty in this research is that one scalar value of $K_n(e)$ is switched with a row vector that can be expressed as $K_n(e) = [K_{n1}(e)K_{n2}(e)K_{n3}(e)]$, which will cause amelioration of the performance of the NPID where the values of nonlinear gains will be tuned based on the error and the type of constant parameters ($K_p$, $K_i$ and $K_d$).

Adaptive control is used for nonlinear systems where some system parameters are unknown, or vary over time. For example, the method known as "Model Reference Adaptive Systems" is one specific approach to solving this issue (MRAS). In order to do this, a reference process model must be defined whose dynamics are in response to a reference input that the plant process should mimic. To obtain the plant's output signal to match the reference model's output signal for the plant process with unknown parameters, a particular control law modifies the reference input signal. In this study, the updated law directly modifies the settings on the NPID controller. It makes the NPID more effective, and insensitive to external disturbances. In addition, it can adapt to violent changes in parameters for the EV system.

Based on the examination of the dynamic characteristics, an improved adaptive control mechanism for the mode transition of a hybrid electric vehicle was demonstrated in [5]. A novel study about a configuration-switchable hydraulically interconnected suspension system under a nonlinear model predictive control was presented in [28]. Dynamic simulation of road/tire longitudinal interaction for designing EV control systems was illustrated in [29]. A lane-keeping control approach with direct yaw moment control input by taking into account the dynamics of an EV was displayed in [30]. A swarm optimization technique-based predictive regenerative braking control approach for a hybrid electric vehicle was executed in [19]. A different design identification and control based on GA optimization for an autonomous wheelchair was developed in [25]. An optimal nonlinear PID speed tracking control based on harmony search (HS) for an EV was used in [31]. A new MRAS for a high-performance pantograph robot mechanism was simulated in [32]. In [33], researchers used diverse machine learning (ML) algorithms for thermal comfort predictive models. Furthermore, in [34], an expert system was built to generate control decision for a maritime transport model. Moreover, [35] illustrated direct torque control of an induction machine using a fuzzy switching controller. The proposed technique achieved satisfactory performance for the induction machine. A novel LabVIEW self-tuning PID controller was implemented in real-time to control the response in [36]. A new algorithm was presented in [37]; the RBFNN (radial basis function neural network) was used in the technique to identify the vehicle's Jacobian information, execute parameter tuning for PID control, and accomplish vehicle longitudinal control with self-tuning capabilities.

This research presents new auto-tuning for the NPID controller based on a model reference adaptive control. The main purpose of the controller was to track a preselected speed profile of the EV with low power. A comparison was conducted between the famous

PID, the conventional NPID, and auto-tuning NPID controllers which were optimized by a new COVID-19 optimization. Several operating points were implemented to validate the controller's performance. Furthermore, the dynamic response of the EV was recorded through the vehicle parameters' uncertainties to ensure the robustness of the proposed controller.

The rest of the study is organized as follows: the second section demonstrates the nonlinear model of an EV; the third section illustrates in detail the design steps for auto-tuning the NPID controller; the fourth section presents the results of the proposed controllers; the last section is the conclusion.

## 2. System Modeling

Propulsion, energy supply, and auxiliary subsystems make up the majority of an EV's three subsystems. According to Figure 1, the propulsion subsystem is made up of the vehicle controller, power electronic converter, electric motor, mechanical transmission, sensors, and driving wheels [38,39]. The energy source, the energy management unit, the charger unit, and other elements are included in the energy supply subsystem [40,41]. The power steering unit, air conditioning motor and its controller, and the auxiliary supply unit make up the auxiliary subsystem. The electric vehicle drivetrain system (EVDS) is created by integrating the subsystems [42,43].

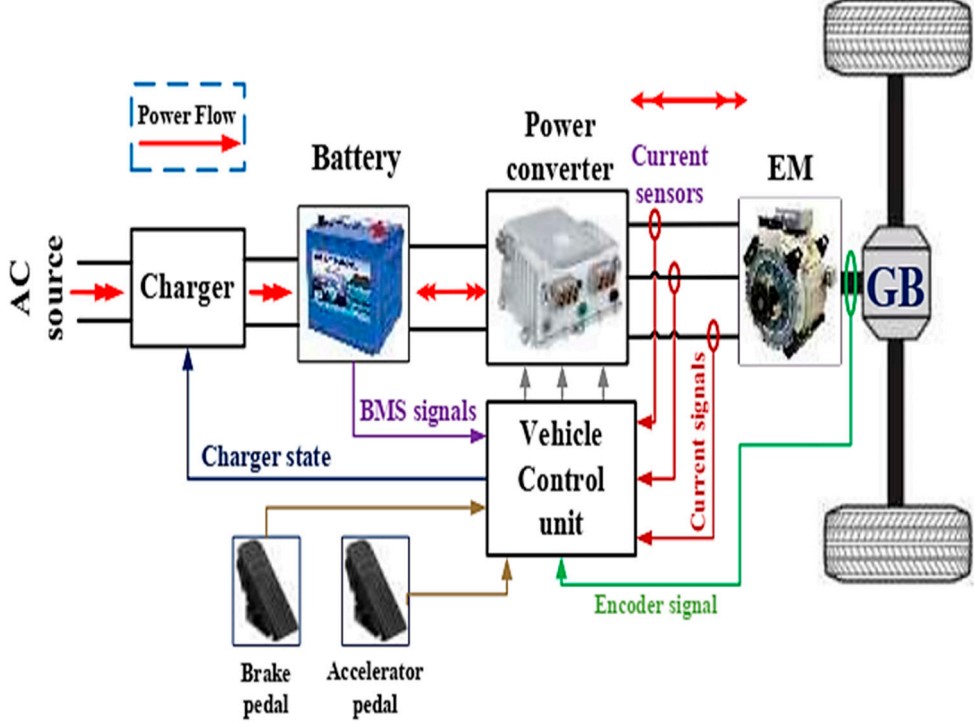

**Figure 1.** Simple structure of the battery and electric vehicle.

The energy source that powers every piece of technology in the car is the battery pack. When the battery is being charged or discharged, the battery management system (BMS) sends voltage, current, and state-of-charge (SOC) signals to the vehicle control unit. The major component of the drivetrain system is the electrical machine (EM). To propel the vehicle with the necessary speed and acceleration, force is produced by the EM in the propulsion systems. The electric motor uses battery power to generate driving force, but it also has the ability to act as a generator when the required deceleration causes the applied reference torque to turn negative.

The EV mainly comprises a battery unit, controller, and electric motors connected to the vehicle through the transmission unit. The EV system dynamics has two parts: vehicle

and motor dynamics. Electric vehicle system modeling involves the balancing of all the forces acting on a running vehicle. There are mainly four types of forces, namely rolling friction ($F_{rr}$), aerodynamic drag force ($F_{ad}$), gravitational force ($F_g$), and force due to vehicle acceleration ($F_a$), as shown in Figure 2. Hence, the total traction force ($F_t$) acting on a vehicle is given by the following:

$$F_t = F_{rr} + F_{ad} + F_g + F_a = \mu_{rr}mg + 0.5\rho AC_d v^2 + mg\sin\varphi + m\, dv/dt \tag{1}$$

where $m$ is the mass of the electric vehicle, $g$ is the gravity acceleration, $v$ is the driving velocity of the vehicle, $\mu_{rr}$ is the rolling resistance coefficient, $\rho$ is the air density, $A$ is the frontal area of the vehicle, $C_d$ is the drag coefficient, and $\varphi$ is the hill-climbing angle.

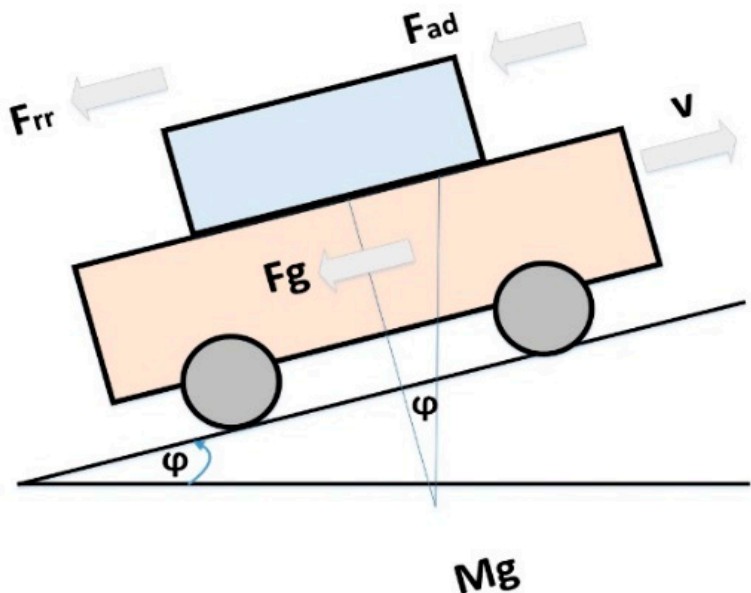

**Figure 2.** Schematic of external obstacles affecting a running EV [7].

The nonlinear state space model of an EV can be considered as follows:

$$\dot{X} = f(X) + g(X)u \tag{2}$$

$$X = \begin{bmatrix} x1 \\ x2 \end{bmatrix} = \begin{bmatrix} i \\ w \end{bmatrix} \tag{3}$$

$$f(x) = \begin{bmatrix} \frac{-(R_a+R_f)}{L_a+L_f}x_1 - \frac{L_{af}}{L_a+L_f}x_1.x_2 \\ \frac{1}{J+m\left(\frac{r^2}{G^2}\right)}\left[ L_{af}x_1^2 - Bx_2 - \frac{r}{G}\left(\mu_{rr}mg + \frac{1}{2}\rho AC_d\left(\frac{r^2}{G^2}\right)x_2^2\right) + mg\sin(\varphi) \right] \end{bmatrix} \tag{4}$$

$$g(x) = \begin{bmatrix} \frac{1}{L_a+L_f} \\ 0 \end{bmatrix} \tag{5}$$

The system parameters are summarized in Table 1.

A nonlinear EV block diagram is presented in Figure 3. The EV system is classified into two main subsystems. The first is the electric motor drive system, and the second is the chassis of the body with suspension parts. The nonlinearity source comes from the mutual inductance of the motor winding; the system parameters variations are shown in Table 2. The purpose of the proposed controllers is to absorb external disturbances such as aerodynamic resistance, road variations, and several operating points of speed; moreover, the controllers overcome internal disturbances such as the mentioned nonlinearity resources, random noise, and uncertainty in the system parameters.

**Table 1.** Parameters of a nonlinear EV system.

| Symbol | Value | Symbol | Value |
|---|---|---|---|
| $L_a + L_f$ | 6.008 mH | $m$ | 800 kg |
| $R_a + R_f$ | 0.12 Ω | $A$ | 1.8 m$^2$ |
| $L_{af}$ | 0.001 mH | $\rho$ | 1.25 (kg/m$^3$) |
| $i$ | 78 A (250 max) | $\varphi$ | 0° |
| $V$ | 0:48 V | $C_d$ | 0.3 |
| $B$ | 0.0002 N.M.s | $\mu_{rr}$ | 0.015 |
| $J$ | 0.05 Kg.m$^2$ | $G$ | 11 |
| $\omega$ | 25 Km/h | r | 0.25 m |

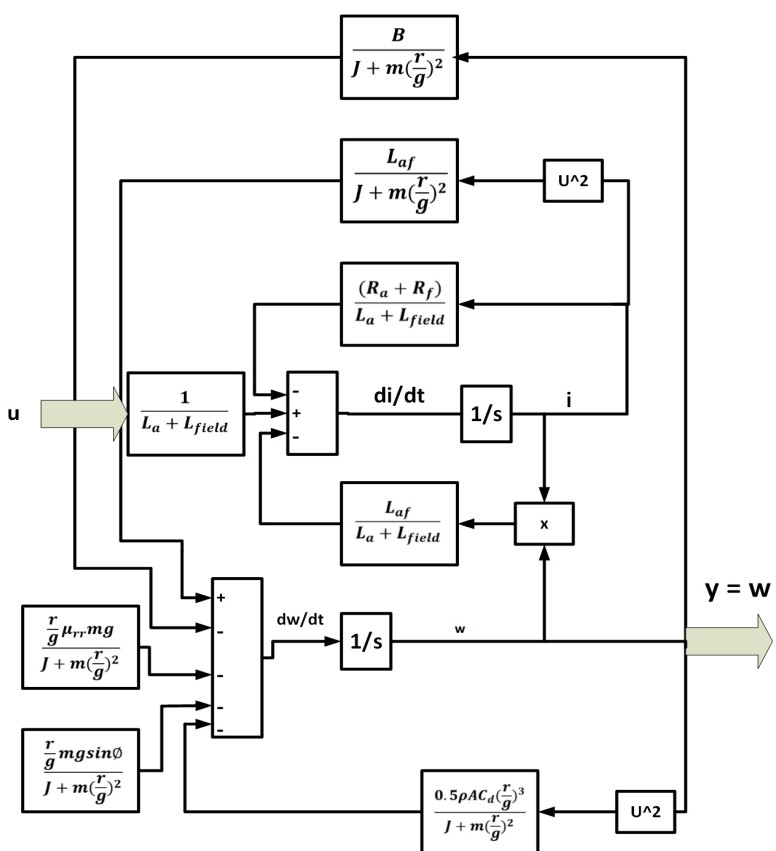

**Figure 3.** SIMULINK diagram of a nonlinear EV model.

**Table 2.** Uncertainty parameters of an EV system.

| Parameter | Variation % |
|---|---|
| $Ra + Rf$ | +10 |
| $La + Lf$ | −20 |
| $r$ | +25 |
| $J$ | −20 |
| $m$ | +30 |
| $Cd$ | −20 |
| $\mu rr$ | +30 |

### 3. Auto-Tuning Nonlinear PID Control

Model reference adaptive control (MRAC) is a high-ranking adaptive controller [22]. It may be regarded as an adaptive servo system in which the desired performance is expressed in terms of a reference model. In this research, the NPID control parameters were adjusted online using the model reference adaptive technique and the nonlinear function.

The proposed form of NLPID control can be described as follows:

$$u(t) = \left(k_p + K_{n1}(e)\right)\left[\, e(t)\right] + (k_i + K_{n2}(e)).\int_0^t \left[\, e(t)\right] dt + (k_d + K_{n3}(e)).\left[\frac{de(t)}{dt}\right] \quad (6)$$

where $K_{n1}(e)$, $K_{n2}(e)$, and $K_{n3}(e)$ are nonlinear gains. The nonlinear gains represent any general nonlinear function in which the error is bounded in the sector $0 < K_n(e) < K_n(e)$ max.

There is a wide range of choices available for the nonlinear gain $K_n(e)$. One simple form of the nonlinear gain function can be described as follows:

$$K_{ni}(e) = ch(w_i e) = \frac{exp(w_i e) + exp(-w_i e)}{2} \quad (7)$$

where $i = 1, 2, 3$.

$$e = \begin{Bmatrix} e & |e| \leq e_{max} \\ e_{max} sgn(e) & |e| > e_{max} \end{Bmatrix}$$

Figure 4 presents the main structure of auto-tuning NPID based on the model reference technique.

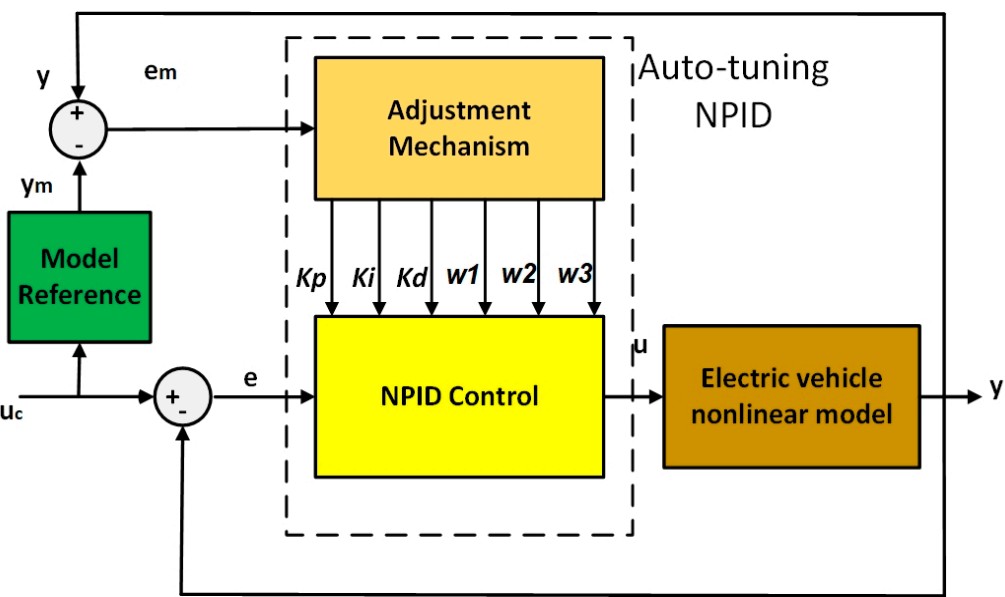

**Figure 4.** The overall system with self-tuning NPID is based on the model reference technique.

The nonlinear gain $K_n(e)$ is lower-bounded by $K_n(e)$ min = 1 when $e = 0$, and upper-bounded by $K_n(e)$ max = ch($w_i e_{max}$). Therefore, $e_{max}$ stands for the range of deviation, and $w_i$ describes the rate of variation of $K_n(e)$.

The MIT rule is the original approach to model reference adaptive control. The name is derived from the fact that it was developed at the Instrumentation Laboratory (now the Draper Laboratory) at MIT. To adjust parameters in such a way that the loss function is minimized, the following is applied:

$$j(\theta) = \frac{1}{2}e_m^2 \quad (8)$$

To make $j$ small, it is reasonable to change the parameters in the direction of the negative gradient of $j$, as follows:

$$\frac{d\underline{\theta}}{dt} = -\gamma \frac{\partial j}{\partial \underline{\theta}} = -\gamma e_m \frac{\partial e_m}{\partial \underline{\theta}} \tag{9}$$

$$e = u_c - y \tag{10}$$

Assuming that the plant can be simplified to a first-order system, this is shown in the following equation:

$$\frac{y(s)}{u(s)} = \frac{k}{Ts + 1} \tag{11}$$

where $k$ and $T$ are unknown parameters. Furthermore, assuming that the model reference takes the form of a first-order system, the following relationship is used:

$$\frac{y_m(s)}{u_c(s)} = \frac{k_m}{T_m s + 1} \tag{12}$$

where $k_m$ and $T_m$ are selected by the designer.

From Equations (10)–(12), the following can be concluded:

$$y = \frac{k}{Ts + 1} \left[ (k_p + K_{n1}) + (k_i + K_{n2}) \frac{1}{s} + (k_d + K_{n3})s \right] (u_c - y) \tag{13}$$

$$y = \frac{k\left[ (k_p + K_{n1}) + (k_i + K_{n2}) \frac{1}{s} + (k_d + K_{n3})s \right]}{Ts + 1} u_c$$
$$- \frac{k\left[ (k_p + K_{n1}) + (k_i + K_{n2}) \frac{1}{s} + (k_d + K_{n3})s \right]}{Ts + 1} y \tag{14}$$

$$\left( 1 + \frac{k\left[ (k_p + K_{n1}) + (k_i + K_{n2}) \frac{1}{s} + (k_d + K_{n3})s \right]}{Ts + 1} \right) y$$
$$= \frac{k\left[ (k_p + K_{n1}) + (k_i + K_{n2}) \frac{1}{s} + (k_d + K_{n3})s \right]}{Ts + 1} u_c$$

$$\left( \frac{Ts + 1 + k\left[ (k_p + K_{n1}) + (k_i + K_{n2}) \frac{1}{s} + (k_d + K_{n3})s \right]}{Ts + 1} \right) y = \frac{k\left[ (k_p + K_{n1}) + (k_i + K_{n2}) \frac{1}{s} + (k_d + K_{n3})s \right]}{Ts + 1} u_c$$

$$y = \frac{k\left[ (k_p + K_{n1}) + (k_i + K_{n2}) \frac{1}{s} + (k_d + K_{n3})s \right]}{Ts + 1 + k\left[ (k_p + K_{n1}) + (k_i + K_{n2}) \frac{1}{s} + (k_d + K_{n3})s \right]} u_c \tag{15}$$

$$e_m = y - y_m \tag{16}$$

$$e_m = \left[ \frac{k\left[ (k_p + K_{n1}) + (k_i + K_{n2}) \frac{1}{s} + (k_d + K_{n3})s \right]}{Ts + 1 + k\left[ (k_p + K_{n1}) + (k_i + K_{n2}) \frac{1}{s} + (k_d + K_{n3})s \right]} - \frac{k_m}{T_m s + 1} \right] u_c \tag{17}$$

$$\frac{\partial e_m}{\partial k_p} = \left[ \frac{(Ts + 1)k}{\left( Ts + k\left[ (k_p + K_{n1}) + (k_i + K_{n2}) \frac{1}{s} + (k_d + K_{n3})s \right] \right)^2} \right] u_c \tag{18}$$

Equation (18) can be rewritten as follows:

$$\frac{\partial e_m}{\partial k_p} = \left[ \frac{(Ts + 1)k}{\left( Ts + k\left[ (k_p + K_{n1}) + (k_i + K_{n2}) \frac{1}{s} + (k_d + K_{n3})s \right] + 1 \right) \left( k\left[ (k_p + K_{n1}) + (k_i + K_{n2}) \frac{1}{s} + (k_d + K_{n3})s \right] \right)} \right] y \tag{19}$$

From Equations (17) and (19), the following is derived:

$$\frac{\partial e_m}{\partial k_p} = \left[ \frac{k^2 e}{\left( Ts + k\left[ (k_p + K_{n1}) + (k_i + K_{n2})\frac{1}{s} + (k_d + K_{n3})s \right] + 1 \right)} \right] \tag{20}$$

To achieve the desired performance, the following condition must hold:

$$Ts + k\left[ (k_p + K_{n1}) + (k_i + K_{n2})\frac{1}{s} + (k_d + K_{n3})s \right] + 1 = T_m s + 1 \tag{21}$$

$$\frac{\partial e_m}{\partial k_p} = \frac{k^2 e}{T_m s + 1} \tag{22}$$

From the MIT rule, one can obtain the following relationship:

$$\frac{dk_p}{dt} = -\gamma.e_m.\frac{k^2 e}{T_m s + 1} \tag{23}$$

$$\frac{dk_p}{dt} = -\gamma_1.\frac{e_m.e}{T_m s + 1} \tag{24}$$

$$\gamma_1 = \gamma.k^2 \tag{25}$$

$$(k_p)_{new} = \int \frac{dk_p}{dt} dt + k_p(0) \tag{26}$$

where $k_p(0)$ is the initial value of proportional gain $k_p$.

$$\frac{\partial e_m}{\partial k_i} = \frac{1}{s}\left[ \frac{k}{Ts + k\left[ (k_p + K_{n1}) + (k_i + K_{n2})\frac{1}{s} + (k_d + K_{n3})s \right] + 1} - \frac{k\left[ (k_p + K_{n1}) + (k_i + K_{n2})\frac{1}{s} + (k_d + K_{n3})s \right]}{\left( Ts + k\left[ (k_p + K_{n1}) + (k_i + K_{n2})\frac{1}{s} + (k_d + K_{n3})s \right] + 1 \right)^2} \right] u_c \tag{27}$$

Equation (28) can be rewritten as follows:

$$\frac{\partial e_m}{\partial k_i} = \frac{1}{s}\left[ \frac{k(Ts + 1)}{\left( Ts + k\left[ (k_p + K_{n1}) + (k_i + K_{n2})\frac{1}{s} + (k_d + K_{n3})s \right] + 1 \right)^2} \right] u_c \tag{28}$$

$$\frac{\partial e_m}{\partial k_i} = \frac{1}{s}\left[ \frac{k(Ts + 1)}{\left( Ts + k\left[ (k_p + K_{n1}) + (k_i + K_{n2})\frac{1}{s} + (k_d + K_{n3})s \right] + 1 \right)\left( k\left[ (k_p + K_{n1}) + (k_i + K_{n2})\frac{1}{s} + (k_d + K_{n3})s \right] \right)} \right] y \tag{29}$$

From Equations (27) and (29), we obtain the following:

$$\frac{\partial e_m}{\partial k_i} = \frac{1}{s}\left[ \frac{k^2 e}{\left( Ts + k\left[ (k_p + K_{n1}) + (k_i + K_{n2})\frac{1}{s} + (k_d + K_{n3})s \right] + 1 \right)} \right] \tag{30}$$

To achieve the desired performance, the condition must hold in Equation (21).

$$\frac{\partial e_m}{\partial k_i} = \frac{1}{s}\frac{k^2 e}{T_m s + 1} \tag{31}$$

From the MIT rule, we can obtain the following relationship:

$$\frac{dk_i}{dt} = -\gamma.e_m.\frac{1}{s}\frac{k^2 e}{T_m s + 1} \tag{32}$$

$$\frac{dk_i}{dt} = -\gamma_2 . \frac{e_m . e}{T_m s + 1} \tag{33}$$

$$\gamma_2 = \gamma k^2 . \frac{1}{s} = \gamma_1 \frac{1}{s} \tag{34}$$

$$(k_i)_{new} = \int \frac{dk_i}{dt} dt + k_i(0) \tag{35}$$

where $k_i(0)$ is the initial value of proportional gain $k_i$.

$$\frac{\partial e_m}{\partial k_d} = \left[ \frac{ks}{Ts + k\left[ (k_p + K_{n1}) + (k_i + K_{n2})\frac{1}{s} + (k_d + K_{n3})s \right] + 1} - \frac{ks\left( k\left[ (k_p + K_{n1}) + (k_i + K_{n2})\frac{1}{s} + (k_d + K_{n3})s \right] \right)}{\left( Ts + k\left[ (k_p + K_{n1}) + (k_i + K_{n2})\frac{1}{s} + (k_d + K_{n3})s \right] + 1 \right)^2} \right] u_c \tag{36}$$

$$\frac{\partial e_m}{\partial k_d} = \left[ \frac{ks^{\mu}\left( Ts + k\left[ (k_p + K_{n1}) + (k_i + K_{n2})\frac{1}{s} + (k_d + K_{n3})s \right] + 1 - k\left[ (k_p + K_{n1}) + (k_i + K_{n2})\frac{1}{s} + (k_d + K_{n3})s \right] \right)}{\left( Ts + k\left[ (k_p + K_{n1}) + (k_i + K_{n2})\frac{1}{s} + (k_d + K_{n3})s \right] + 1 \right)^2} \right] u_c \tag{37}$$

$$\frac{\partial e_m}{\partial k_d} = \left[ \frac{ks(Ts + 1)}{\left( Ts + k\left[ (k_p + K_{n1}) + (k_i + K_{n2})\frac{1}{s} + (k_d + K_{n3})s \right] + 1 \right)^2} \right] u_c \tag{38}$$

$$\frac{\partial e_m}{\partial k_d} = \left[ \frac{ks(Ts + 1)}{\left( Ts + k\left[ (k_p + K_{n1}) + (k_i + K_{n2})\frac{1}{s} + (k_d + K_{n3})s \right] + 1 \right)\left( k\left[ (k_p + K_{n1}) + (k_i + K_{n2})\frac{1}{s} + (k_d + K_{n3})s \right] \right)} \right] y \tag{39}$$

Furthermore, from Equations (43) and (45), we obtain the following:

$$\frac{\partial e_m}{\partial k_d} = \left[ \frac{k^2 . s . e}{\left( Ts + k\left[ (k_p + K_{n1}) + (k_i + K_{n2})\frac{1}{s} + (k_d + K_{n3})s \right] + 1 \right)} \right] \tag{40}$$

$$\frac{\partial e_m}{\partial k_d} = \frac{k^2 . s . e}{T_m s + 1} \tag{41}$$

$$\frac{dk_d}{dt} = -\gamma . e_m . \frac{k^2 . s . e}{T_m s + 1} \tag{42}$$

$$\frac{dk_d}{dt} = -\gamma_3 . \frac{e_m . e}{T_m s + 1} \tag{43}$$

$$\gamma_3 = \gamma . k^2 . s = \gamma_1 . s \tag{44}$$

$$(k_d)_{new} = \int \frac{dk_d}{dt} dt + k_d(0) \tag{45}$$

where $k_d(0)$ is the initial value of the derivative gain $k_d$.

There are many methods to define the parameters of the control techniques, such as trial and error and Ziegler–Nichols methods for PID control; however, most of these methods are rough roads [44,45]. This study used a new effective optimization technique, which is the COVID-19 optimization algorithm, to find the optimal parameters of the proposed controllers (PID and NPID) based on the output response behavior and the desired performance, as in [17].

The initial population contains the upper and lower values for each control technique. The performance of each row was investigated according to the objective function in Equation (6). Poor performance specifies the infected population, which has the possi-

bility to die; meanwhile, good performance indicates the recovered population from the corona virus:

$$\delta_t = (\delta_1 + \delta_2 + \delta_3 + \delta_4)/4 \tag{46}$$

$$\delta_1 = \frac{|t_r - t_{rd}|}{t_{rd}} \tag{47}$$

$$\delta_2 = \frac{|t_s - t_{sd}|}{t_{sd}} \tag{48}$$

$$\delta_3 = \frac{|e_{ss} - e_{ssd}|}{e_{ssd}} \tag{49}$$

$$\delta_4 = \frac{|OS - OS_d|}{OS_d} \tag{50}$$

where $(t_{rd})$ is the desired rise time and $(t_r)$ is the measured rise time; $(O_{sd})$ is the desired maximum overshoot and $(OS)$ is the actual overshoot; $(t_{sd})$ is the desired settling time and $(t_s$ is the determining settling time; and $(e_{ssd})$ is the desired steady-state error and $(e_{ss})$ is the estimated steady-state error.

It should be observed that the objective function uses four sub-objective functions to try to appease the designer. Improving the rising time of all drive systems is the first sub-objective function. Reduced settling time is the second sub-objective function. The steady-state error is measured by the third sub-objective function. The needed overshoot is investigated in the fourth sub-objective function. Each sub-objective function has a value between zero and one. Therefore, the total objective function takes into account the average of the sum of the four sub-goal functions.

The obtained controller parameters are shown in Table 3.

**Table 3.** Uncertainty parameters of the EV system.

| Controller Type | Parameter | Value |
|:---:|:---:|:---:|
| PID control | $k_p$ | 5.254 |
| | $k_i$ | 0.05 |
| | $k_d$ | 0.02 |
| NPID Control | $k_p$ | 10.23 |
| | $k_i$ | 2.23 |
| | $k_d$ | 1.58 |
| | $w_1$ | 0.25 |
| | $w_2$ | 0.34 |
| | $w_3$ | 0.01 |
| Auto-Tuning NPID Control | $k_p(0)$ | 6.35 |
| | $k_i(0)$ | 0.125 |
| | $k_d(0)$ | 2.31 |
| | $w_1(0)$ | 0.45 |
| | $w_2(0)$ | 0.69 |
| | $w_3(0)$ | 0.78 |

## 4. Results and Discussion

This section illustrates the proposed controller's performance for the EV under external and internal disturbances. Several tests were executed to validate the robustness and flexibility of the proposed controllers. The first test used a single operating speed to measure the dynamic response of each control technique.

Figure 5 demonstrates the closed-loop dynamic behavior of the EV through a single operating speed. It can be noted that the auto-tuning NPID controller had a smooth response, low rise time and settling time, and no overshoot, which made the EV comfortable. In contrast, the other techniques (PID and NPID controllers) required a long time to stabilize the desired operating point. Moreover, they had a high overshoot, which made the EV's motion unstable.

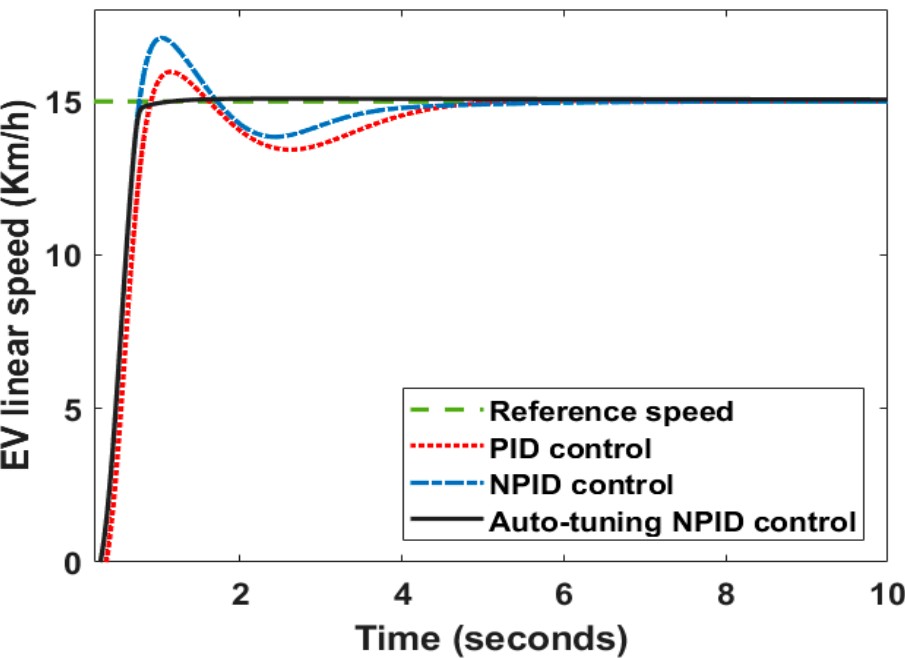

**Figure 5.** EV linear speed response through the proposed control techniques.

The reasons for the poor performance of these controllers (PID and NPID controllers) were the inability to overcome system uncertainty, and the internal and external disturbances. Moreover, these controllers without adaptive mechanisms integrated into the case of the auto-tuning NPID controller made the parameters have to continuously adapt with the system's sudden changes.

Figure 6 demonstrates the rotor winding current in amperes (A) of each control technique. It can be noted that the starting current was identical for the proposed controllers, while the current stabilized quickly in the case of the auto-tuning NPID controller. Both the PID and NPID had high fluctuations in current.

Figure 7 demonstrates the performance of the PID, NPID, and auto-tuning NPID to track the New European Drive Cycle (NEDC) speed (Km/h) test. It is obvious that the auto-tuning NPID controller tracked the speed profile accurately. Furthermore, the PID and the NPID controllers could not track the continuous changes in the speed profile. A zoomed area was taken from 980 s to 1020 s to ensure that the auto-tuning NPID had a small settling time and smooth behavior compared to other control techniques that did not have an adaptive mechanism.

Figure 8 demonstrates the corresponding rotor current in amperes (A) for the NEDC test. It was demonstrated that the PID and NPID controllers had a high starting current at each change in speed profile, which made them consume a lot of power. In contrast, the auto-tuning NPID controller had a low starting current compared to the other control techniques (PID and NPID), which saved power consumption and permitted the EV to move longer than for the other techniques.

Figure 9 shows vehicle velocity, in km/h, based on the proposed control techniques within the Urban Dynamometer Driving Schedule (UDDS) driving cycle for the same time interval.

It was noticed that the auto-tuning NPID controller accurately tracked its reference velocity, although there were violent changes in the reference speed and nonlinearity resources of the EV. In the case of the PID and NPID controllers, they could not track the profile where the gap between the reference and the actual velocity was high. A zoomed area demonstrated the difference between the reference and the actual velocity. It can be noted that the error was 6.307 minus 6.271, equaling 0.036.

Figure 10 illustrates the corresponding instantons current in amperes (A) for the UDDS test. It was noted that the PID and NPID controllers had a high value of starting current at each change in operating point, while the auto-tuning NPID controller had a low peak current compared to the other control techniques. This meant that the auto-tuning NPID controller consumed low power throughout the driving cycle. Moreover, Table 4 demonstrates that the auto-tuning NPID controller had the least peak and average current throughout this cycle.

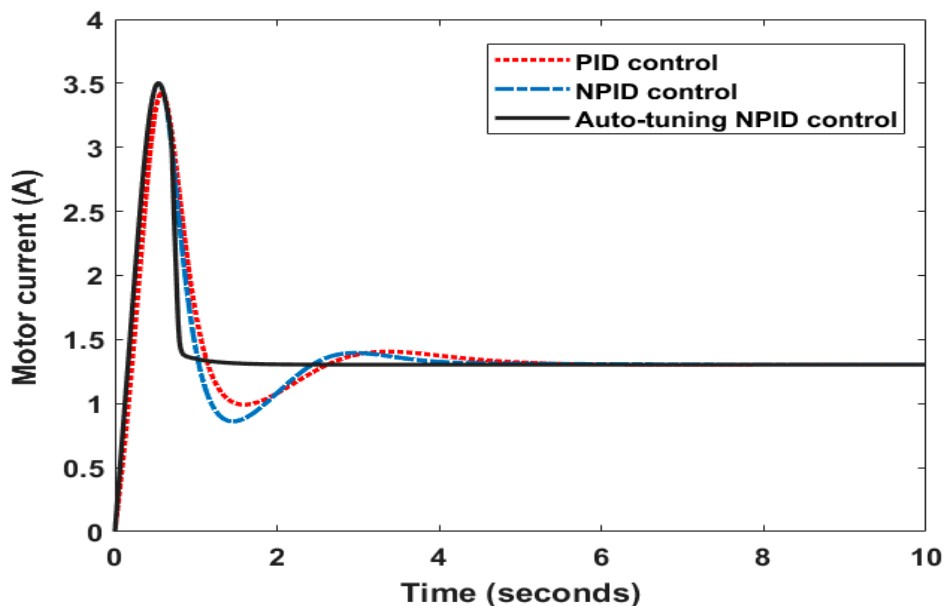

**Figure 6.** The corresponding rotor current of the motor through a fixed speed reference.

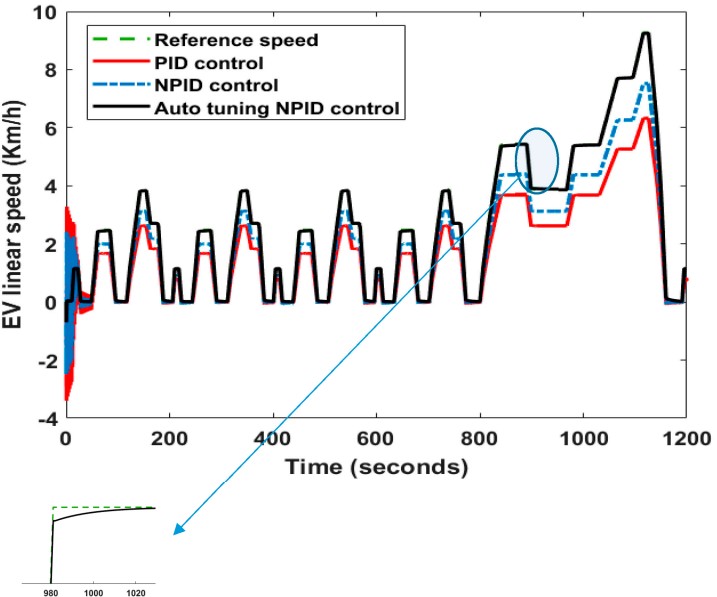

**Figure 7.** Performance of the PID, NPID, and auto-tuning NPID to track the NEDC speed test.

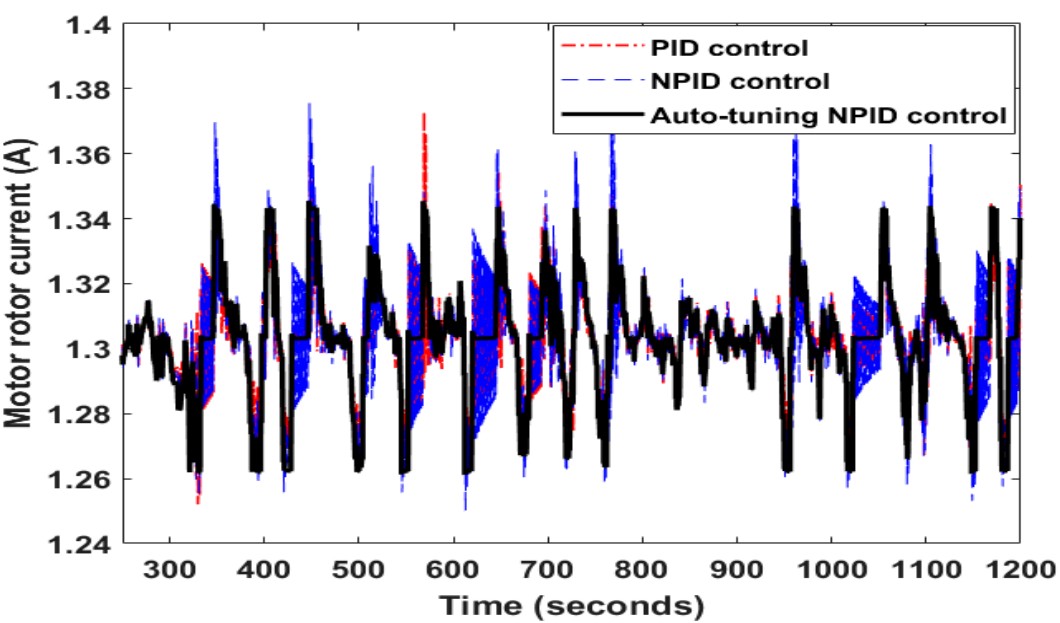

**Figure 8.** The corresponding currents of the PID, NPID, and auto-tuning NPID to track the NEDC speed test.

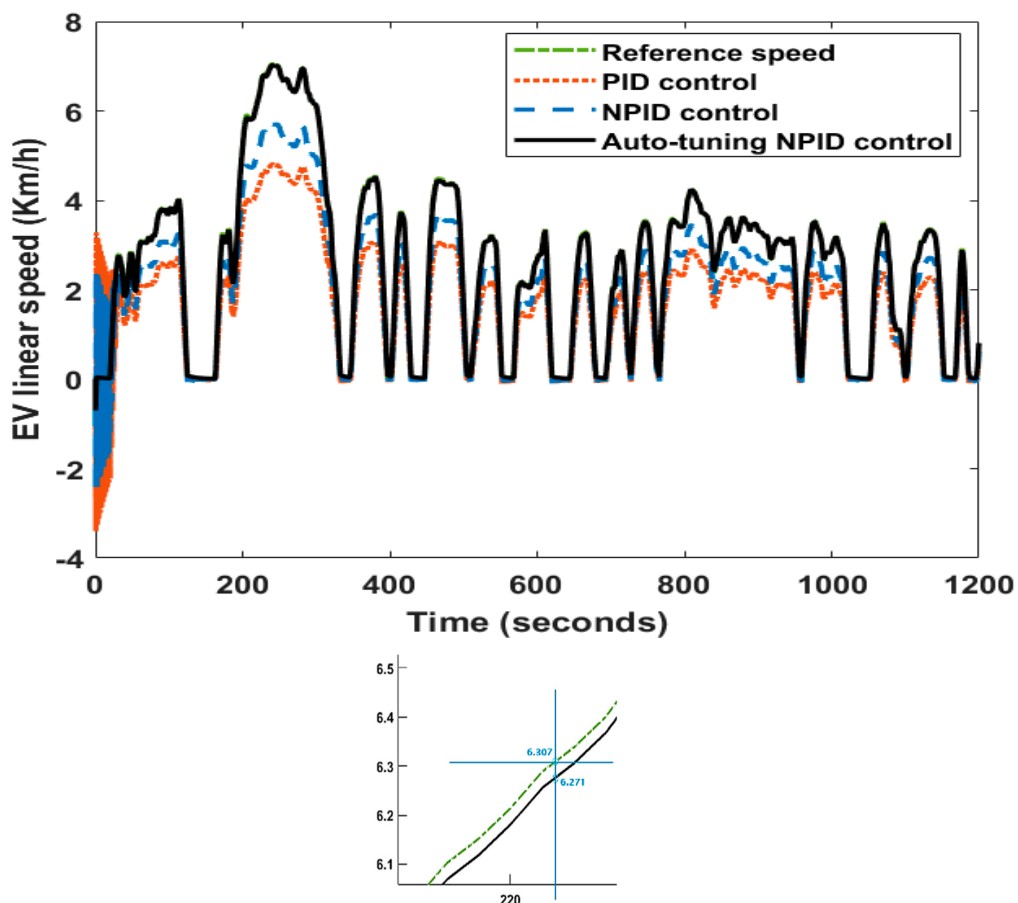

**Figure 9.** Performance of the PID, NPID, and auto-tuning NPID to track the UDDS test.

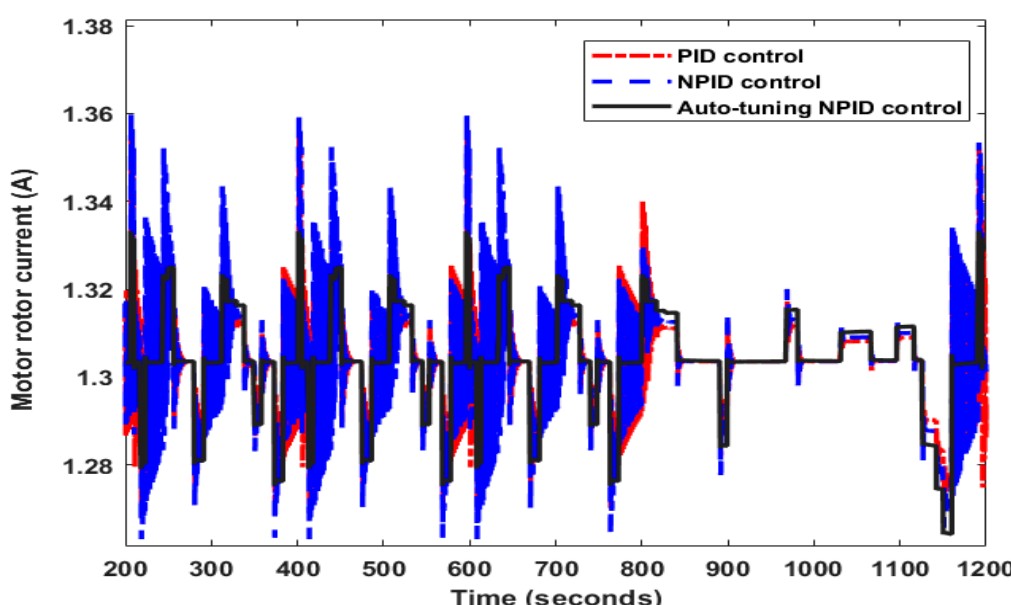

**Figure 10.** The corresponding instantons current in amperes (A) for the UDDS test.

**Table 4.** The current consumption for each control technique.

| Controller Type | Peak Current (A) | Average Current (A) |
|---|---|---|
| PID controller | 1.35 | 0.912 |
| NPID controller | 1.36 | 0.851 |
| Auto-tuning NPID controller | 1.33 | 0.712 |

Table 4 demonstrates a comparison to show the consumed current for each control technique. It can be noted that the proposed auto-tuning NPID controller used the least current by assuming that the output voltage of the EV battery was constant.

## 5. Conclusions

In this study, a new auto-tuning nonlinear PID controller was introduced for an electric vehicle (EV) model. Two objectives were intended to be achieved by the recommended control. The main objective was to enhance the EV's dynamic response to both internal and external shocks. Reducing power consumption for EVs was the second objective. Two well-known controllers were installed to ensure that these objectives would be met by evaluating the dynamic performance and power consumption. The first was a PID controller that used COVID-19 optimization. The second employed a nonlinear PID (NPID) controller, which was COVID-19-tuned. The results demonstrate that, in comparison to other control systems, the auto-tuning NPID had a comfortable dynamic response and a short rise and settling time (PID and NPID controllers). Additionally, via the driving cycles, it achieved a low continuous power (peak current 1.33 A and average current 0.712 A).

**Funding:** This research received no external funding.

**Data Availability Statement:** Not applicable.

**Conflicts of Interest:** The authors declare no conflict of interest.

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
