# Peer review of "Design of Auto-Tuning Nonlinear PID Tracking Speed Control for Electric Vehicle with Uncertainty Consideration"

_wevj, doi:10.3390/wevj14040078_

Round 1

Reviewer 1 Report

Below is a list of the parts that can be improved/ that are not clear:

1) Check figures 7, 8, 9 -> label y axis missing

2) Standardize equations' font 

3) Please provide further detail on the optimization procedure implemented along with related literature

4) Please provide further comments about the novelty of the work

5) The author didn't cite any work related to auto-tuning procedure. An analysis of the literature in this regard could be useful to improve the methodology and obtain higher performances

6) Quite concerned about performance in the first seconds of the driving mission (see figure 7, 8, 9). Why this behaviour cannot be improved? 

7)  Non-exhaustive conclusion; they should be rewritten

8) The author wrote "Also, it achieves a low continuous power through the driving cycles" -> please provide detailed discussion and evaluation

9) The novelty contribution should be further discussed 

10) Several typos detected (A general reading of the paper should be done prior to submission)

Author Response

Dear Reviewers and Editors,

We would like to use this opportunity to sincerely thank the reviewers for their interest in our work and for helpful comments that will greatly improve the manuscript. As indicated below, we have checked all the comments provided by the Referees and have made necessary changes.

In the following answers, all reviewers’ comments are in black, and our answers are in blue.

Best Regards,

Mohamed A. Shamseldin

1) Check figures 7, 8, 9 -> label y axis missing

Done

2) Standardize equations' font 

Done

3) Please provide further detail on the optimization procedure implemented along with related literature

Done

4) Please provide further comments about the novelty of the work

Done

5) The author didn't cite any work related to auto-tuning procedure. An analysis of the literature in this regard could be useful to improve the methodology and obtain higher performances

Done

6) Quite concerned about performance in the first seconds of the driving mission (see figure 7, 8, 9). Why this behaviour cannot be improved? 

7)  Non-exhaustive conclusion; they should be rewritten

Done

8) The author wrote "Also, it achieves a low continuous power through the driving cycles" -> please provide detailed discussion and evaluation

Done (table 4)

9) The novelty contribution should be further discussed 

Done

10) Several typos detected (A general reading of the paper should be done prior to submission)

Done

Reviewer 2 Report

Article is interesting, the subject is current and has useful value. In order to enhance the article quality, I suggest the following remarks be taken into account:

1.       “However, AI-based controllers suffer from draw-backs, such as large data requirements, extended learning, and training duration.” Please delete the above sentence. It seems to be a little too strong a statement. It is good idea to add a few references, that refer to AI-based controllers in various applications, for instance:

-   Jack Ngarambe, Geun Young Yun, Mat Santamouris, The use of artificial intelligence (AI) methods in the prediction of thermal comfort in buildings: energy implications of AI-based thermal comfort controls, Energy and Buildings, Volume 211, 2020, 109807

-   Borkowski P. „Inference engine in an intelligent ship course-keeping system” Computational Intelligence and Neuroscience vol. 2017, art. no. 2561383, 2017

-   Hanumanthakari, S. & Kodad, S.F. & Sarvesh, B. (2016). Improved Fuzzy Logic based DTC of Induction machine for wide range of speed control using AI based controllers. Journal of Electrical Systems, vol. 12, 301-314.

2.       Please mark the vectors in bold.

3.       Equation 26: Incorrect mathematical notation.

4.       Table 3: The authors should add units.

5.       The authors are suggested to have Discussion section to investigate the weakness, strength, and potential enhancement of proposed scheme.

6.       It is good idea to add a few sentences about future analysis.

Author Response

Dear Reviewers and Editors,

We would like to use this opportunity to sincerely thank the reviewers for their interest in our work and for helpful comments that will greatly improve the manuscript. As indicated below, we have checked all the comments provided by the Referees and have made necessary changes.

In the following answers, all reviewers’ comments are in black, and our answers are in blue.

Best Regards,

Mohamed A. Shamseldin

Article is interesting, the subject is current and has useful value. In order to enhance the article quality, I suggest the following remarks be taken into account:

  1. “However, AI-based controllers suffer from draw-backs, such as large data requirements, extended learning, and training duration.” Please delete the above sentence. It seems to be a little too strong a statement. It is good idea to add a few references, that refer to AI-based controllers in various applications, for instance:

-   Jack Ngarambe, Geun Young Yun, Mat Santamouris, The use of artificial intelligence (AI) methods in the prediction of thermal comfort in buildings: energy implications of AI-based thermal comfort controls, Energy and Buildings, Volume 211, 2020, 109807

-   Borkowski P. „Inference engine in an intelligent ship course-keeping system” Computational Intelligence and Neuroscience vol. 2017, art. no. 2561383, 2017

-   Hanumanthakari, S. & Kodad, S.F. & Sarvesh, B. (2016). Improved Fuzzy Logic based DTC of Induction machine for wide range of speed control using AI based controllers. Journal of Electrical Systems, vol. 12, 301-314.

Done

  1. Please mark the vectors in bold.

Done

  1. Equation 26: Incorrect mathematical notation.

Equation 26 describes the adaptation law of parameter.

  1. Table 3: The authors should add units.

The table 3 contain the controller parameters which are without units

  1. The authors are suggested to have Discussion section to investigate the weakness, strength, and potential enhancement of proposed scheme.

Done

  1. It is good idea to add a few sentences about future analysis.

Done

Reviewer 3 Report

Authors have presented the manuscript in an innovative manner. The method and results are acceptable.

1] Authors may verify the references cited in the introduction and their relevance.

2] literatures can be presented in a tabular format.

3] The presented Fig.1 is taken from any literature or its a own figure.

4] Justification of fig.2

5] what is the relevance of equation 4 and 5.

6] mathematical analysis is acceptable in this explanation.

7] any significance  of proposed NPID controller ?

8] figure 7 can be improved ?

Author Response

Dear Reviewers and Editors,

We would like to use this opportunity to sincerely thank the reviewers for their interest in our work and for helpful comments that will greatly improve the manuscript. As indicated below, we have checked all the comments provided by the Referees and have made necessary changes.

In the following answers, all reviewers’ comments are in black, and our answers are in blue.

Best Regards,

Mohamed A. Shamseldin

Authors have presented the manuscript in an innovative manner. The method and results are acceptable.

1] Authors may verify the references cited in the introduction and their relevance.

Done

2] Literatures can be presented in a tabular format.

Done

3] The presented Fig.1 is taken from any literature or its a own figure.

The figure redrawn and added some features from ref [7].

4] Justification of fig.2

Done

5] what is the relevance of equation 4 and 5.

Equations (4) and (5) demonstrate the variables f(x) and g(x) of equation (2)

6] mathematical analysis is acceptable in this explanation.

Done

7] any significance  of proposed NPID controller ?

The NPID control without online tuning used to compare to the new proposed auto-tuning NPID controller.

8] figure 7 can be improved ?

Done

Reviewer 4 Report

In the proposed manuscript, the Author declares that he has developed a new self-tuning non-linear PID controller for a non-linear electric vehicle model. Thanks to this, according to the author, it is possible to improve the dynamic performance of the electric vehicle (EV) and minimize the energy consumption of the EV. Are you sure the Author is right?

1. Provide more appealing title with no acronyms in precise and concise manner.

2. Avoid trivial information.

3. Explain in brief how the present paper differs from the published ones. What are the main achievements of the author against the current state of science?

4. No explanations for the abbreviations used. A table with a description of all abbreviations should be included

5. Provide better quality Figures 5-10.

- the axes in the drawings are not described, there are no units, the fonts are too large in relation to the publication text, etc.

- Fig.5: why is the speed less than zero in the beginning of the waveform?

6. Not all physical parameters of the modeled vehicle and drive system are given (currents, voltages, powers, torques, energy storage capacity, gear ratios, etc.)

7. The COVID-19 optimization method, which is used as the main parameter optimization method, is not discussed. It is not appropriate in this situation to indicate "and COVID-19 optimization [14-17]."

8. Please correct Fig.4 and describe all markings shown on it.

9. On what basis were the parameters presented in table 3 determined?

10. As specific goals, the author, through the use of the developed regulator, assumed the improvement of vehicle dynamics and the reduction of energy consumption. According to the reviewer, the goals set were not proven by the author. There are no unambiguous results, e.g. expressed as a percentage of how the vehicle dynamics changed (no reference to the regulation quality indicators). Lack of proof of a strong thesis that energy consumption has been reduced. By what percent has energy consumption been reduced? From the presented results, it can be clearly observed that for the adopted UDDS and NEDC test cycles, the set value of the vehicle speed was not reached. For the author's information: the test is considered positive if the difference between the set speed and the speed obtained by the vehicle is not less than 2%. Reducing the speed of the vehicle during the test is an obvious cause of the reduction in energy consumption, which the author cannot claim as an achievement.

11. What is "a new auto-tuning nonlinear PID controller for a nonlinear Electric Vehicle (EV) model" is not explained.

12. Get its English edited very carefully.

13. State main findings in the conclusions.

Author Response

Dear Reviewers and Editors,

We would like to use this opportunity to sincerely thank the reviewers for their interest in our work and for helpful comments that will greatly improve the manuscript. As indicated below, we have checked all the comments provided by the Referees and have made necessary changes.

In the following answers, all reviewers’ comments are in black, and our answers are in blue.

Best Regards,

Mohamed A. Shamseldin

In the proposed manuscript, the Author declares that he has developed a new self-tuning non-linear PID controller for a non-linear electric vehicle model. Thanks to this, according to the author, it is possible to improve the dynamic performance of the electric vehicle (EV) and minimize the energy consumption of the EV. Are you sure the Author is right?

  1. Provide more appealing title with no acronyms in precise and concise manner.

Done

  1. Avoid trivial information.

Done

  1. Explain in brief how the present paper differs from the published ones. What are the main achievements of the author against the current state of science?

Done

  1. No explanations for the abbreviations used. A table with a description of all abbreviations should be included

Done

  1. Provide better quality Figures 5-10.

Done

- the axes in the drawings are not described, there are no units, the fonts are too large in relation to the publication text, etc.

Done

- Fig.5: why is the speed less than zero in the beginning of the waveform?

You are right but it depends on the MATLAB solver type and initial condition and revised to begin from zero.

  1. Not all physical parameters of the modeled vehicle and drive system are given (currents, voltages, powers, torques, energy storage capacity, gear ratios, etc.)

Symbol

Value

Symbol

value

+

6.008 mH

800 kg

+

0.12

1.8 m2

0.001 mH

1.25 (kg/m3)

78 A(250 max)

0

0:48 V

0.3

0.0002 N.M.s

0.015

0.05 Kg.m2

11

25 Km/h

r

0.25 m

  1. The COVID-19 optimization method, which is used as the main parameter optimization method, is not discussed. It is not appropriate in this situation to indicate "and COVID-19 optimization [14-17]."

Done

  1. Please correct Fig.4 and describe all markings shown on it.

Done

  1. On what basis were the parameters presented in table 3 determined?

Based on the COVID-19 Optimization.

  1. As specific goals, the author, through the use of the developed regulator, assumed the improvement of vehicle dynamics and the reduction of energy consumption. According to the reviewer, the goals set were not proven by the author. There are no unambiguous results, e.g. expressed as a percentage of how the vehicle dynamics changed (no reference to the regulation quality indicators). Lack of proof of a strong thesis that energy consumption has been reduced. By what percent has energy consumption been reduced? From the presented results, it can be clearly observed that for the adopted UDDS and NEDC test cycles, the set value of the vehicle speed was not reached. For the author's information: the test is considered positive if the difference between the set speed and the speed obtained by the vehicle is not less than 2%. Reducing the speed of the vehicle during the test is an obvious cause of the reduction in energy consumption, which the author cannot claim as an achievement.

     Table 4 demonstrates the consumed current from the battery at a fixed voltage for each control technique. It can be noted that the auto-tuning NPID control has the least average current through this test which makes the EV save power and also, it has a good dynamic response.

  1. What is "a new auto-tuning nonlinear PID controller for a nonlinear Electric Vehicle (EV) model" is not explained.

This paper presents a new auto-tuning for the NPID controller based on a model reference adaptive control. The main purpose of the controller is to track a preselected speed profile of the EV with low power. A comparison between the famous PID, and conventional NPID and auto-tuning NPID controllers which are optimized by a new COVID-19 optimization. Several operating points had been implemented to validate the controller performance. Also, the dynamic response of the EV is recorded through the vehicle parameters uncertainty to ensure the robustness of the proposed controller.

  1. Get its English edited very carefully.

Done

  1. State main findings in the conclusions.

Done

Round 2

Reviewer 1 Report

The authors didnt answer to: Quite concerned about performance in the first seconds of the driving mission (see figure 7, 8, 9). Why this behaviour cannot be improved? 

Conclusions and english can be improved. 

Author Response

Dear Reviewers and Editors,

We would like to use this opportunity to sincerely thank the reviewers for their interest in our work and for helpful comments that will greatly improve the manuscript. As indicated below, we have checked all the comments provided by the Referees and have made the necessary changes.

In the following answers, all reviewers’ comments are in black, and our answers are in blue.

Best Regards,

Mohamed A. Shamseldin

6) Quite concerned about performance in the first seconds of the driving mission (see figures 7, 8, 9). Why this behaviour cannot be improved? 

Figures 7 & 9in the first seconds are demonstrated below. It can be noted that the auto-tuning track accurately the profile speed in contrast to the other techniques.

Reviewer 2 Report

I accept the amendments made by authors. From my side the work is accepted in this new version.

Author Response

Thank you for your kind response 

Reviewer 4 Report

The corrections presented by the author are cosmetic, and in addition, the author writes that some elements have been "done" and nothing has been corrected, eg ad.5.

Where is it noted that a comment has been corrected? The word "done" doesn't say that. this is why the lines in the manuscript are numbered so that the author can indicate the places of his corrections.

In addition, the reviewer once again draws attention to the fact of scientific incompatibility. The test is considered completed if the set value in the tests differs from the one obtained by a maximum of 2%. So how can the author compare his results where the differences are much larger and draw wrong conclusions on this basis.

WEV is a serious journal and a certain scientific standard should be maintained.

Author Response

Dear Reviewers and Editors,

We would like to use this opportunity to sincerely thank the reviewers for their interest in our work and for helpful comments that will greatly improve the manuscript. As indicated below, we have checked all the comments provided by the Referees and have made necessary changes.

In the following answers, all reviewers’ comments are in black, and our answers are in blue.

Best Regards,

Mohamed A. Shamseldin

In the proposed manuscript, the Author declares that he has developed a new self-tuning non-linear PID controller for a non-linear electric vehicle model. Thanks to this, according to the author, it is possible to improve the dynamic performance of the electric vehicle (EV) and minimize the energy consumption of the EV. Are you sure the Author is right?

In this paper, the proposed technique makes the EV moves smoothly and accurately. Also, it consumed a low current compared to the other control technique as demonstrated in the results by assuming a constant battery voltage.  

  1. Provide more appealing title with no acronyms in precise and concise manner.

The only abbreviation of the paper title is PID which is familiar in the control system field.

  1. Avoid trivial information.

All mentioned information is important to be the paper clear for readers), please definite the type of trivial information you mean.

  1. Explain in brief how the present paper differs from the published ones. What are the main achievements of the author against the current state of science?

The paper presents a new approach to adapt the traditional nonlinear PID control to enhance the EV dynamic performance under uncertain conditions. Also, the proposed technique saves the consumed power of the battery. Moreover, a new optimization technique was used to determine the proposed control technique. 

  1. No explanations for the abbreviations used. A table with a description of all abbreviations should be included

All abbreviations are mentioned throughout the paragraphs. It is difficult and not professional to make a table for abbreviations.

  1. Provide better quality Figures 5-10.

All figures in the paper redrawn

- The axes in the drawings are not described, there are no units, the fonts are too large in relation to the publication text, etc.

The axes in the drawings are modified for figures 7,8 and 9

- Fig.5: why is the speed less than zero in the beginning of the waveform?

You are right but it depends on the MATLAB solver type and initial condition and revised to begin from zero.

  1. Not all physical parameters of the modeled vehicle and drive system are given (currents, voltages, powers, torques, energy storage capacity, gear ratios, etc.)

Symbol

Value

Symbol

value

+

6.008 mH

800 kg

+

0.12

1.8 m2

0.001 mH

1.25 (kg/m3)

78 A(250 max)

0

0:48 V

0.3

0.0002 N.M.s

0.015

0.05 Kg.m2

11

25 Km/h

r

0.25 m

  1. The COVID-19 optimization method, which is used as the main parameter optimization method, is not discussed. It is not appropriate in this situation to indicate "and COVID-19 optimization [14-17]."

Done (from sentence 306 to 326)

  1. Please correct Fig.4 and describe all markings shown on it.

Done

  1. On what basis were the parameters presented in table 3 determined?

Based on the COVID-19 Optimization.

  1. As specific goals, the author, through the use of the developed regulator, assumed the improvement of vehicle dynamics and the reduction of energy consumption. According to the reviewer, the goals set were not proven by the author. There are no unambiguous results, e.g. expressed as a percentage of how the vehicle dynamics changed (no reference to the regulation quality indicators). Lack of proof of a strong thesis that energy consumption has been reduced. By what percent has energy consumption been reduced? From the presented results, it can be clearly observed that for the adopted UDDS and NEDC test cycles, the set value of the vehicle speed was not reached. For the author's information: the test is considered positive if the difference between the set speed and the speed obtained by the vehicle is not less than 2%. Reducing the speed of the vehicle during the test is an obvious cause of the reduction in energy consumption, which the author cannot claim as an achievement.

     Table 4 demonstrates the consumed current from the battery at a fixed voltage for each control technique. It can be noted that the auto-tuning NPID control has the least average current through this test which makes the EV save power and also, it has a good dynamic response.

  1. What is "a new auto-tuning nonlinear PID controller for a nonlinear Electric Vehicle (EV) model" is not explained.

This paper presents a new auto-tuning for the NPID controller based on a model reference adaptive control. The main purpose of the controller is to track a preselected speed profile of the EV with low power. A comparison between the famous PID, conventional NPID, and auto-tuning NPID controllers which are optimized by a new COVID-19 optimization. Several operating points had been implemented to validate the controller's performance. Also, the dynamic response of the EV is recorded through the vehicle parameters uncertainty to ensure the robustness of the proposed controller.

  1. Get its English edited very carefully.

Done (the paper was checked by a grammar checker)

  1. State main findings in the conclusions.

Done

Round 3

Reviewer 1 Report

none

Author Response

Thanks for your kind response

Reviewer 4 Report

The reviewer also pointed out that the model of the vehicle and the symbols used are not sufficiently explained.

The presented results do not at all confirm the thesis presented by the author.

The reviewer once again draws attention to the fact of scientific incompatibility. The test is considered completed if the set value in the tests differs from the one obtained by a maximum of 2%. How can the author compare his results where the differences are much larger and draw wrong conclusions on this basis? 

Author Response

Dear Reviewers and Editors,

We would like to use this opportunity to sincerely thank the reviewers for their interest in our work and for helpful comments that will greatly improve the manuscript. As indicated below, we have checked all the comments provided by the Referees and have made necessary changes.

In the following answers, all reviewers’ comments are in black, and our answers are in blue.

Best Regards,

Mohamed A. Shamseldin

The reviewer also pointed out that the model of the vehicle and the symbols used are not sufficiently explained.

All available parameters for EV added in the paper.

The presented results do not at all confirm the thesis presented by the author.

The results enhanced by table 4 and more explanation

The reviewer once again draws attention to the fact of scientific incompatibility. The test is considered completed if the set value in the tests differs from the one obtained by a maximum of 2%. How can the author compare his results where the differences are much larger and draw wrong conclusions on this basis?

Each test investigates the proposed controllers at several conditions to ensure the robustness of the controller. It can achieve high performance for some tests and satisfactory performance for another test but overall the proposed controller has excellent performance compared to known used controllers( PID, NPID).
